# Recent Advances and Challenges in Management of *Colletotrichum orbiculare*, the Causal Agent of Watermelon Anthracnose

Takshay Patel [1], Lina M. Quesada-Ocampo [2], Todd C. Wehner [1,*], Bed Prakash Bhatta [3,4], Edgar Correa [5] and Subas Malla [3,4,*]

1   Department of Horticultural Science, North Carolina State University, Raleigh, NC 27695, USA; takshaypatel8@gmail.com
2   Department of Entomology and Plant Pathology, North Carolina State University, Raleigh, NC 27695, USA; lmquesad@ncsu.edu
3   Department of Horticultural Sciences, Texas A&M University, College Station, TX 77843, USA; bedprakash.bhatta@ag.tamu.edu
4   Texas A&M AgriLife Research and Extension Center, Uvalde, TX 78801, USA
5   Department of Soil and Crop Sciences, Texas A&M University, College Station, TX 77843, USA; edgar.correa@tamu.edu
*   Correspondence: tcwehner@gmail.com (T.C.W.); subas.malla@ag.tamu.edu (S.M.)

**Abstract:** The fungus *Colletotrichum orbiculare* causes watermelon anthracnose and is an important pathogen of watermelon in the United States, causing a significant impact on yield and quality of the produce. The application of fungicides as preventative and post-occurrence control measures is currently being deployed by growers. Further study of the genetic and molecular basis of anthracnose resistance will help in guiding future watermelon breeding strategies. Several conserved virulence factors (effectors) in *C. orbiculare* have been reported to interact with the host, at times impairing the host immune machinery. A single dominant gene conferring race 1 anthracnose resistance was reported independently on two watermelon germplasm. The recent advances in genomics, transcriptomics, proteomics, and metabolomics could facilitate a better understanding of the interaction between *C. orbiculare* effectors and host resistance genes in the already sequenced watermelon genome. In this review, we encompass and discuss (i) the history of watermelon anthracnose, taxonomy, morphology, and diversity in races of *C. orbiculare*; (ii) the epidemiology of the anthracnose disease and host resistance; (iii) the genetics behind the pathogenesis; and (iv) the current advances in breeding and molecular efforts to elucidate anthracnose resistance.

**Keywords:** anthracnose; *Colletotrichum orbiculare*; watermelon



## 1. Introduction

Watermelon is an important crop grown worldwide with 102 Megatons produced in 2021 [1]. In the United States (U.S.), the crop had a total economic value of USD 748 million in 2022 [2]. Watermelons are grown in most U.S. states, but the majority of the production occurs in Florida, Georgia, California, and Texas because of higher temperatures and a longer growing season [3]. The introduction of seedless cultivars has increased the per capita consumption of watermelon by 37% since 1980 [3]. As the watermelon industry grew, challenges related to fruit quality, yield, and production methods emerged. As is the case with many crops, disease and pest management is a significant limitation to watermelon production. Diseases are a major priority for watermelon producers since new races of pathogens continue to break down host resistance and develop insensitivity to fungicides. The major diseases of watermelons are Fusarium wilt, anthracnose, gummy stem blight, powdery mildew, Phytophthora, downy mildew, bacterial fruit blotch, and viruses. While Fusarium wilt and Phytophthora are considered the most devastating

diseases to watermelon when fields become infested, foliar diseases such as anthracnose, gummy stem blight, downy mildew, and powdery mildew affect the crop on a yearly basis, forcing growers to make significant investments in cultural practices and crop protection to manage these diseases [4].

## 2. History of Watermelon Anthracnose

Anthracnose has been a major problem in watermelon production worldwide at least since the 19th century, and it occurs wherever cucurbits are grown in a humid climate. Passerini in Italy first observed anthracnose on calabash/bottle gourd in 1867 [5]. In 1875, Passerini reported anthracnose on watermelon and cantaloupe, which is the first known scientific report of anthracnose on watermelon. More reports came from Europe during the late 19th century. In the U.S., Dr. Eckfeldt (Philadelphia) and Prof. A. B. Seymour (Wisconsin) noted anthracnose on gourds and watermelons, respectively, in 1885. In 1889, Galloway reported melon anthracnose in New Jersey, Virginia, and North Carolina. Substantial losses of cantaloupes, cucumbers, and watermelons due to anthracnose epidemics started in 1904 in Nebraska, Indiana, New Jersey, West Virginia, North Carolina, South Carolina, Wisconsin, and Ohio. Yield losses of up to 30% in watermelons [6] and 60% in cucurbits [7] have been reported to be caused by anthracnose. Anthracnose became a major plant disease during the late 19th century, and by the early 20th century, many U.S. states started focusing on anthracnose as an important watermelon pathogen [8].

## 3. The Pathogen

### 3.1. The Causal Agent: Colletotrichum Orbiculare

The genus *Colletotrichum* is economically and scientifically important because it contains many plant-pathogenic fungi infecting a wide range of crops; those crops include row crops, fruits, flowers, and vegetables. Almost every domesticated crop is a host of a species belonging to the genus *Colletotrichum* [9]. In many hosts, *Colletotrichum* spp. infect all the aerial parts of the plant, including stems, leaves, fruits, and flowers. *Colletotrichum* spp. are also significant postharvest pathogens since spores from foliar infections during field growth can cause an infection that progresses during transport or on the market shelves, resulting in complete loss of the crop [10]. When the plant pathogens were defined based on host specificity, *Colletotrichum* consisted of almost 700 species. Later, von Arx reclassified those species in 1957 to 11 taxa based on morphological traits [11,12]. *Colletotrichum* has been used as a model system for biochemical, physiological, and genetic studies. The concept of pathogen races was initially recognized in *Colletotrichum lindemuthianum* [13]. Many of the studies that laid the foundation for the concept of systemic acquired resistance were performed using the cucumber model of *Colletotrichum orbiculare* [14].

*Colletotrichum orbiculare* ((Berk. & Mont.) Arx) is an important pathogen of cucurbits including cucumber, muskmelon, watermelon, squash, gourd, pumpkin, cantaloupe, honeydew, and *Luffa* spp. [15]. *C. orbiculare* can also infect tobacco [16]. Anthracnose can cause severe damage, both in the field and postharvest, and is one of the top research priorities for watermelon in the U.S. [17]. In the field, *C. orbiculare* infects all the above-ground parts of plants, including leaves, stems, flowers, and fruits [3,5,8]. Infections on any of these plant parts have direct effects on yield. The pathogen also infects all growth stages of the plant, from the seedling stage to mature fruit-bearing plants. Defoliation will leave the plant with poor photosynthetic capacity, stunt fruit development, and expose the mature fruit to direct sunlight, leading to sunburn. Infection on both growing and mature plants leads to unmarketable produce.

### 3.2. Taxonomy

*Colletotrichum orbiculare* belongs to the kingdom Fungi, phylum Ascomycota, class Sordariomycetes, order Glomerellales, family Glomerellaceae, genus *Colletotrichum* [18,19].

Earlier when the taxonomic classification of plant pathogenic fungi was based on plant disease specificity, *C. orbiculare* was named and identified multiple times by different

researchers around the world [12]. Cucurbit and bean anthracnose were assumed to be caused by the same fungus, which was named *Colletotrichum lagenarium* [8]. This assumption was discarded in a comparative study of anthracnose fungi, in which bean anthracnose was named *Volutella citrulli*. Based on modern molecular tests, *C. orbiculare* is recognized as a species complex, with *C. lindemuthianum*, *C. malvarum*, *C. orbiculare*, and *C. trifolii* as distinct species [20]. Currently, the isolates causing watermelon anthracnose are classified as a subspecies in the *C. orbiculare* species group. Researchers still differ in classifying this pathogen [18].

### 3.3. Fungal Morphology

The mycelium at first is colorless, thin-walled, septate, and uniformly cylindrical. Many of the cells later increase in diameter up to threefold, becoming thick-walled and dark brown in color [8,18]. On culture media, mycelium is first colorless and then pink and black at the end. Pink coloration is sometimes observed in host tissues, with blackening of mycelium being common in fruit lesions. Acervuli, anthracnose mycelium aggregates, branch and become intertwined to send out a layer of short colorless conidiophores [8]. From the tip of the conidiophores, spores bud off apically one at a time, piling up to form a pink slimy cluster on top of acervuli. Spores are surrounded by a sticky water-soluble matrix and are single-celled, clear, oblong or ovate, and vaguely pointed at one end [8]. Spore size varies from 13 to 19 μm by 4 to 6 μm, and masses are pink in color. Acervuli have two to three long setae scattered among the conidiophores, which are brown, thick-walled bristles 90 to 120 μm in length [8,18]. The number of setae in a single acervulus varies and can be up to 36. Spores form heaps as high as setae, with setae supporting the spore mass. Sclerotial bodies are typically observed more on media and fruit lesions and are formed due to the further development of the bases of stromata or acervuli, where the whole mass is enlarged and black in color [8]. On media, the spore mass may dry and remain as a part of sclerotial bodies, whereas in fruit lesions, the spores are washed away, and the black stroma that forms the black spots on fruits remains [8].

Germinating spores form an appressorium at the tip of each of the germ tubes, which are brown, melanized, thick-walled, and ovoid to spherical in shape. Appressoria are slightly tapered at one end and flattened on the side of contact with the host [8]. Melanization of appressoria is important for pathogenicity. Melanin-deficient mutants have reduced pathogenicity, and melanization is supposed to resist the high turgor pressure within the appressorium and direct the force on the leaf epidermis for penetration [21]. Melanization in fungal spores is also assumed to provide protection under adverse conditions like oxygen radicals, high temperatures, irradiation, or lysis by other microbes [22].

### 3.4. Life and Disease Cycle

*Colletotrichum orbiculare* is the asexual form of the cucurbit anthracnose pathogen and propagates through conidia. It normally exits in the asexual stage, and it rarely undergoes the sexual stage [18,23]. There has been no defined complete life cycle for *C. orbiculare* and only a few reports of the sexual stage of *C. orbiculare*. The sexual stage of *C. orbiculare* was reported as a species of *Glomerella* but was not classified [23]. Ascospores are produced in abundance when paired with other isolates of *C. orbiculare*, but few ascospores develop when isolates are selfed [23].

There is a close relation between *C. orbiculare* spread and wet weather conditions like rain, morning dew, and overhead irrigation. Conidia are mainly dispersed by rain splashing, but also by wind, instruments, and workers [4]. Spore heaps in acervuli are surrounded by a sticky water-soluble matrix [8,18], explaining the need for moisture for spread. The importance of moisture for *C. orbiculare* was observed and established in the early 20th century when anthracnose epidemics were new. A wetness period of 16 h or more shows maximum disease development [24]. Further, temperatures from 18 °C to 27 °C (65° to 80 °F) are ideal for the establishment and growth of *C. orbiculare* on watermelon [25]. *C. orbiculare* overwinters by surviving on the debris of infected plants.

Cucurbit anthracnose was more severe on fields that had melons as previous crops [26]. Sheldon documented the spread of *C. orbiculare* by the transportation of diseased fruit and contaminated seeds [26]. Overall, *C. orbiculare* spreads by rain, irrigation, seeds, fruit, and overwintering and survives between seasons on infected plant debris, on volunteer plants, and in and on seeds from infected fruits [4].

### 3.5. Infection Process

*Colletotrichum orbiculare* is a hemibiotrophic fungus; during the initial stage of infection, it behaves as a biotrophic pathogen, keeping the host cells alive, and later it takes nutrients from dead host cells, switching to the necrotrophic stage [27]. *C. orbiculare* penetrates host leaves using two entry modes: turgor-mediated invasion (TMI) via melanized appressoria and hyphal tip-based entry (HTE). During TMI, *C. orbiculare* penetrates the adaxial epidermis [8]. After landing, the spores adhere to the plant surface and then germinate to produce germ tubes and further form melanized appressoria. The appressoria penetrate plant epidermal cells directly through the cuticle and cell wall but not from the stomata [28]. The epidermal cell wall below the appressorium swells, mostly due to cell-wall-degrading enzymes secreted by *C. orbiculare* [27,28]. After penetration, biotrophic intracellular hyphae develop inside the host cells, infecting via intracellular colonization at the cellular level. The intracellular hypha is surrounded by an intact host cell plasma membrane, growing within the plant cell lumen, i.e., between the plant plasma membrane and plant cell walls [29]. The infection then proceeds to the necrotrophic phase where the secondary necrotrophic hyphae arise from the intracellular hyphae, obtaining nutrients from dead host cells [27,28]. HTE works independently of the melanized appressoria and is a morphogenic response at wound sites [30]. The existence of these two invasive strategies implies a sensing system that induces the respective morphogenesis response on wound sites and intact leaf tissue for pathogenesis.

In watermelon fruits, the hyphae grow throughout the rind, and acervuli are formed after 4 to 5 days of infection. Conidiophores form conidia masses rupturing the rind epidermis. In resistant watermelon plants, the appressorium entry during foliar disease is the same as in a susceptible plant but the hyphae are only able to infect a few cells around the penetration site [28]. Plant cells around infected leaf sites elongate, divide, and form a raised compact mass to resist fungal growth [28], most likely through lignification. Fruits from resistant plants develop raised areas that are greener as compared to the surrounding rind and remain darker even when the remaining rind starts to bleach [28]. Like what is seen in leaves of resistant plants, *C. orbiculare* only infects one to two epidermal cells in the fruit rind after penetration [28].

### 3.6. Disease Symptoms

*Colletotrichum orbiculare* causes anthracnose in all cucurbits, and the symptoms on each of the species vary. All the above-ground parts of plants are susceptible to anthracnose. Photosynthetic cells are more susceptible than non-photosynthetic tissue [28]. Lesions gradually increase in size with abundant acervuli formation (acervuli are conidiophores producing mycelium aggregates). The descriptions of symptom were added later [4]. On watermelon leaves, anthracnose produces blackish-brown lesions (Figure 1A). Centers of older lesions on leaves fall out, giving it a 'shot hole' appearance. Petioles and stems show sunken and dark-color spindle-shaped lesions, which penetrate deeply and finally grid the stem (Figure 1D). Infected young fruits show aborted growth or are abnormal. Lesions on young fruit are small, black depressed spots. On mature fruits, lesions start as yellow translucent centered elevated pimples, which later turn into flat-topped, circular, water-soaked elevations (Figure 1B,C). Lesions on mature fruit further sink and show pink spore masses on a black or cream-colored background. The black lesions are the result of the black stroma left behind after the washing of spores, whereas the pink masses are like the spore masses found in culture media.

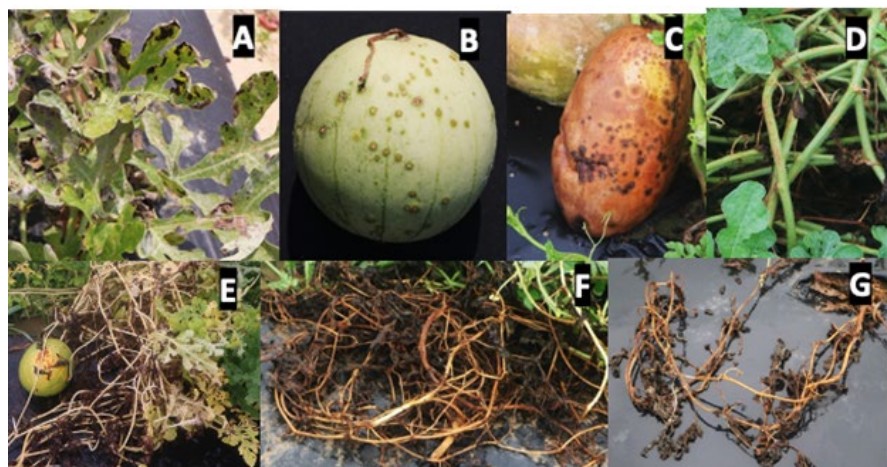

**Figure 1.** Anthracnose symptoms on watermelon. (**A**) Leaf; (**B**,**C**) fruit; (**D**) stem; (**E**–**G**) foliage.

### 3.7. Pathogenesis Genes and Effectors

The average genome size of *Colletotrichum* species is 40 Mb, but *C. orbiculare* has a surprisingly large genome of 90 Mb [27]. *C. orbiculare* expresses a large arsenal of genes during the infection process, including 287 protease-encoding genes, 327 plant cell-wall-degrading enzymes, 700 small secreted proteins (SSPs), and many secondary metabolite backbone-forming proteins [27]. All of these are expressed at a higher level than in other species such as *C. graminicola* and *C. higginsianum*. SSPs and secondary metabolite synthesis genes are upregulated during the initial biotrophic stage of infection, whereas degrading enzymes are upregulated during the later necrotrophic stage of infection. The upregulation of SSP genes during early infection in *C. orbiculare* suggests their importance in maintaining biotrophy during infection. As *C. orbiculare* is a hemibiotroph, there is an orchestrated expression of the effector genes during the shift from biotrophy to necrotrophy [31].

Although through genomic and transcriptomic studies it is known that *C. orbiculare* expresses an arsenal of genes involved in pathogenesis, only a few effectors and pathogenic pathways are known yet. *C. orbiculare* produces several effectors, which include necrosis- and ethylene-inducing peptide1-like protein1 (NLP1) [32,33], DN3, necrosis-inducing secreted protein1 (NIS1) [34,35], MC69 [36], and suppression of immunity in *Nicotiana benthamiana* (SIB1, SIB2) [37]. NLP1 is expressed specifically in necrotrophic invasive hyphae [33], has cytotoxic activity, induces cell death, and triggers immune response [38]. The effector DN3 of *C. orbiculare* suppresses NLP1-triggered plant cell death [39], which was also previously observed in the *C. higginsianum* DN3 homolog [40]. NIS1 induces cell death and is expressed in bulbous biotrophic primary hyphae, but its activity reduces in necrotrophic hyphae. Homologs of NIS1 are present in *C. higginsianum*, *C. graminicola*, and *Magnaporthe oryzae*, suggesting it is a conserved sequence in *Colletotrichum* species. Although NIS1-knockout mutants are virulent on tobacco [35], the transgenic expression of NIS1 in Arabidopsis made the plant susceptible to *C. orbiculare* [41]. Intuitively, the expression of cell-death-inducing NIS1 during the biotrophic phase suggests it is a recognized avirulence (AVR) protein. Recently, it has been shown that NIS1 associates with the receptor-like kinases (RLKs) such as brassinosteroid insensitive 1 (BRI1)-associated kinase (BAK1) and receptor-like cytoplasmic kinases (RLCKs) such as botrytis-induced kinase1 (BIK1) in the host, inhibits pathogen-associated molecular pattern (PAMP)-induced reactive oxygen species (ROS) generation, and ultimately impairs pathogen-associated molecular pattern (PAMP)-triggered immunity (PTI) [42]. DN3 suppresses NIS1-triggered HR-like cell death [35]. MC69 is one of the characterized effectors in *C. orbiculare*. Although the MC69 mutants of *C. orbiculare* had normal colony morphology and conidiogenesis, they had reduced lesion development on cucumber and tobacco [36]. *C. orbiculare* expresses MC69 predominantly during the biotrophic phase of infection. A recent study has proposed two

novel effectors of *C. orbiculare* (SIB1 and SIB2), where SIB1 was found to be expressed at both the primary and later stages of host invasion in tobacco and cucumber, respectively [37].

Few signaling infection-related morphogenesis and pathogenesis pathways have been identified in *C. orbiculare*. *C. orbiculare* has three cascades important for virulence: CMK1, MAF1, and RPK1. The CMK1 cascade is involved in conidial germination, infection, growth, and appressorium maturation. The MAF1 cascade is required for appressorium differentiation, whereas the RPK1 pathway is essential for vegetative growth and conidiation. The melanization of appressoria is important for the normal function of *C. orbiculare* [21], and three melanin biosynthesis enzyme genes, PKS1, SCD1, and THR1, and one regulatory gene, CMR1, have been characterized [43–47]. Mutants of melanin-related genes showed defects in the melanization of appressoria and penetration ability. Further, fatty acid oxidation of peroxisomes is also required for melanization and metabolic processes involved in turgor generation for penetration [41].

Plants have receptors that recognize PAMPs. An example of a PAMP is chitin, a major cell wall component in filamentous fungi. *C. orbiculare* has the SSD1 gene involved in cell wall integrity. Mutants of *ssd1* were not virulent and induced host defense response along with papillae formation. The *ssd1* mutants had increased induction of salicylic acid-induced protein kinase (SIPK) and wound-induced protein kinase (WIPK) activity as compared to the wild type. The effector secretion mechanism has also been studied. *C. orbiculare* shows a strong ring-like signal of the effectors around the primary biotrophic hyphae. The ring signal is present in the interfacial region between the host and *C. orbiculare*, and not inside *C. orbiculare* [31,33]. *C. orbiculare* continuously secretes effectors towards the interfacial region. Effectors are also secreted in a single-dot fashion at the bottom of the appressoria and at putative interfaces on the hyphal surface.

### 3.8. Pathogen Races

The first commercial anthracnose-resistant watermelon cultivars were released in 1949 and were widely used to manage the rampant anthracnose outbreaks at that time. Since then, *C. orbiculare* has emerged and overcome host resistance. In 1958, severe anthracnose symptoms on resistant watermelon cultivars were observed in North Carolina [48]. Pathogen isolates from the new symptoms were observed to be undistinguishable from earlier *C. orbiculare* isolates. Isolates were defined into races based on the differential host reactions on different cucurbit cultivars. Most isolates before 1954 were defined as race 1. Isolates found in 1954 and 1955 were defined as race 2 and were highly pathogenic on all cucurbit cultivars of that time. Some isolates were defined as race 3. The difference between race 1 and race 3 was pathogenicity on squash cultivars. The new isolates had no morphological and cultural differences and were considered as different races [48]. Race 4 [49] and races 5, 6, and 7 [50] were also identified later.

Overall, from 1954 to 1964, seven pathological races of *C. orbiculare* were identified based on differential host reactions. The seven races were based on virulence differences on different cucurbit species and cultivars. Race diversity was reevaluated, and the seven races of *C. orbiculare* were combined into three vegetative compatibility groups (VCGs) by using 92 isolates from the U.S. [15]. Based on vegetative compatibility (a phenomenon where fungi with certain genetic similarity can fuse together to form a single heterokaryon), 11 VCGs were formed among the 92 *C. orbiculare* isolates. Out of the 11 VCGs, only 3 were pathogenic on cucurbits, namely VCGs 1, 2, and 3. VCGs 1 and 3 showed virulence on similar hosts. The watermelon cultivar 'Charleston Gray' was resistant to both VCGs 1 and 3, but susceptible to VCG 2. Similar resistance was shown by cucumber cultivars 'Poinsett 76' and 'Gy 14'. VCGs 1 and 3 were classified as race 1 virulence phenotype, and VCG 2 as race 2 virulence phenotype. A race 2B has also been somewhat characterized through vegetative compatibility and virulence [51]. Race 2B has been found on watermelon, bottle gourd, and muskmelon and belongs to VCG 2 [4].

## 4. Disease Management

Field management practices for anthracnose include planting disease-free seed material, deep plowing of crop residue immediately after harvest, crop rotation with non-cucurbit crops for a minimum of 1 year (2 to 3 years is optimum), avoiding usage of farm machinery among fields when the foliage is wet, fungicide applications with effective active ingredients such as quinone outside inhibitors (QoIs), and resistant plant cultivars [4]. To reduce fruit damage by anthracnose, growers are recommended to avoid mechanical injury to fruits, inspect for infected fruits during harvest and discard them, disinfect the fruit surface with chlorinated water, and refrigerate the fruit after harvest to prevent or delay anthracnose development postharvest [4].

### 4.1. Host Resistance

Host resistance screening was conducted either on cotyledons [15] or the two- to four-true-leaf stage [52–57]. The disease inoculum used for the seedling screening varied from $2.5 \times 10^3$ [57] to $5 \times 10^5$ [54] conidial spores/mL. The incubation time also varied from 3 to 14 days post-inoculation (DPI). A recent study reported the optimum inoculum concentration and DPI for screening two- to four-leaf-stage seedlings for race 2 anthracnose resistance [53]. The authors of this study found that $1 \times 10^5$ spores/mL and a single disease rating of percent leaf lesion at 9 DPI were optimal.

In 1937, Layton started breeding for anthracnose resistance and identified sources of resistance to develop commercial cultivars. Five cultivars from Africa with high resistance to anthracnose were identified, out of which three had edible fruit and desired horticultural traits [5]. The three cultivars were named Africa 8, 9, and 13, and were further used as parents. Homozygous anthracnose-resistant selections from Africa 8, 9, and 13 were crossed with commercial cultivars Iowa Belle and Iowa King and a few other cultivars [5]. The commercial Iowa cultivars were wilt-resistant, large-fruited, and crisp-fleshed. The first widely accepted anthracnose-resistant watermelon cultivars were 'Congo' (1949), 'Fairfax' (1953), and 'Charleston Gray' (1955), released by Andrus [57,58]. Charleston Gray, Congo, and Fairfax are resistant to races 1 and 3 but susceptible to race 2 [48]. Cultivars resistant to race 1 were also resistant to race 3 [57,59].

Resistance to race 2 was first found in a citron, W695, which was also resistant to races 1 and 3 [57]. PI 326515 was the first PI reported to have resistance only to race 2 [56]. More resistance sources to race 2 including PI 189225, 271775, 271778, and 512385 were identified [52,55]. Resistance to anthracnose race 2 was also identified in *Citrullus colocynthis*, designated as R309 [60]. Interestingly, two studies found that resistance in *Citrullus colocynthis*, R309, did not follow the single gene inheritance and was suggested to be multigenic [60]. These studies suggested that a dominant single gene confers major resistance, but there are other genes contributing to the phenotype. R309 has been the only source of multigenic resistance; no other multigenic resistance sources have been reported.

The first inheritance work on anthracnose resistance was performed in 1937 [5]. Resistance to race 1 is dominant over susceptibility and segregates as a single gene. Resistance to races 1 and 3 is controlled by the same gene, *Ar-1* [57]. Inheritance of race 2 resistance is like race 1 resistance, dominant and segregating as a single gene [56]. In Korea, Jang et al. used a biparental population, 'DrHS7250' (female parent, resistant breeding line) and 'Oto9491' (male parent, susceptible breeding line), to identify a *C. orbiculare* race 1-resistance quantitative trait locus (QTL) on chromosome 8 and further conducted transcriptomics via RNAseq on the parents to identify a coiled-coil (CC)–nucleotide-binding site (NBS)–leucine-rich repeat (LRR) gene in the QTL region that conferred resistance to the disease [61]. They hypothesized that residue 18 of a conserved motif, IxxLPxSxxxLYNLQTLxL, could govern resistance in 'DrHs7250'. An independent study conducted in the U.S. using a biparental mapping population, 'Charleston Gray' (female parent, resistant) and 'New Hampshire Midget' (male parent, susceptible), found a major *C. orbiculare* race 1-resistance QTL in the same region on chromosome 8 [62]. A PACE SNP marker designed from the SNP marker CL 14-27-9, identified earlier [61], was also the diagnostic marker for the QTL (LOD = 14.06)

in the study. Even though the resistance source for the breeding line 'DrHS7250' was not reported, it seemed that both 'DrHS7250' and 'Charleston Gray' may have the same resistance gene for *C. orbiculare* race 1. Both studies in different watermelon resistance sources validated that the race 1 anthracnose resistance is governed by a single dominant gene. Even today, anthracnose is a problem and a major research priority in watermelon [17]. Most of the current commercial cultivars with anthracnose resistance were developed by private industry (Table 1). These commercial cultivars have intermediate to high levels of resistance to anthracnose race 1, and some descriptions do not specify the race. Many hybrid watermelon cultivars are resistant to races 1 and 2B and susceptible to race 2 [4]. The SNP marker CL 14-27-9 could be utilized as a diagnostic marker to develop race 1-resistant cultivars via marker-assisted selection in watermelon breeding programs.

**Table 1.** Anthracnose-resistant cultivars of watermelon.

| Cultivar | Level of Resistance | Race | Company |
|---|---|---|---|
| SSX8585 | High | 1 | Sakata, Yokohama, Japan |
| Valentino | High | 1 | Sakata, Yokohama, Japan |
| Belmont | Intermediate | 1 | Sakata, Yokohama, Japan |
| Sweet Treasure | Intermediate | 1 | Sakata, Yokohama, Japan |
| Fascination | Intermediate | 1 | Syngenta, Basel, Switzerland |
| Melody | Intermediate | 1 | Syngenta, Basel, Switzerland |
| Excursion | Intermediate | 1 | Syngenta, Basel, Switzerland |
| Captivation | Intermediate | 1 | Syngenta, Basel, Switzerland |
| Cooperstown | High | 1 | Seminis, St. Louis, MO, USA |
| Majestic | High | ? [†] | Seminis, St. Louis, MO, USA |
| Road Trip | High | ? | Seminis, St. Louis, MO, USA |
| Santa Matilde | High | 1 | Seminis, St. Louis, MO, USA |
| HMX 1925 | Intermediate | 1 | HM Clause, Davis, CA, USA |
| Maistros F1 | High | 1 | HM Clause, Davis, CA, USA |
| Accomplice | High | 1 | HM Clause, Davis, CA, USA |
| Millennium | High | 1 | HM Clause, Davis, CA, USA |

[†] Resistance to anthracnose race was not specified in the varietal description by the company.

*4.2. Crop Protection*

Growers often use fungicides to manage watermelon anthracnose throughout the growing season. Fungicides can be applied preventatively if cost-effective, or application should be started with the occurrence of the symptoms in a 5-to-10-day interval. If disease severity is high or environmental conditions are conducive to disease (wet weather), growers will use the shorter application interval. Effective fungicide active ingredients for managing watermelon anthracnose include compounds in group 11: trifloxystrobin, azoxystrobin, pyraclostrobin, fluoxastrobin; group 7: boscalid, fluxapyroxad; group 3: difenoconazole; group M05: chlorothalonil; and group M03: mancozeb [63,64]. Group 11 fungicides correspond to quinone outside inhibitors (Q$_o$Is), group 7 fungicides correspond to succinate dehydrogenase inhibitors (SDHIs), group 3 fungicides correspond to demethylation inhibitors (DMIs), and M05 and M03 have multi-site contact activity. Products commonly recommended for watermelon anthracnose control include 'Kocide 3000' (Copper Hydroxide), 'Pristine' (pyraclostrobin, boscalid), 'Cabrio' (pyraclostrobin), 'Quadris Top' (azoxystrobin, difenoconazole), 'Bravo WeatherStik' (chlorothalonil), and 'TopGuard EQ' (azoxystrobin) [63,65–67].

**5. Prospects and Challenges**

Although anthracnose has been an important watermelon disease for around a century, there are still many unanswered questions regarding pathogen biology and disease management. Anthracnose races have been identified for over 60 years now, but the genetic bases of those races remain unknown, forcing researchers to rely on differential phenotypic responses for race identification. In genomic studies like that of [27] the genome

of *C. orbiculare* was sequenced, comparative studies were conducted, and transcriptomic analysis unraveling candidate effectors involved during different stages of infection was performed. Recently, the core effector NIS1 was shown to attack the conserved immune kinases in the host system and disrupt the first layer of plant immunity (PTI) [42]. We are yet to understand how plant resistance genes interact with the *C. orbiculare* effectors through the second layer of defense, i.e., effector-triggered immunity (ETI).

Many genomic resources have been developed for cucurbit crops and watermelon in particular [68–70]. Nonetheless, genetic determinants of anthracnose resistance have not been clearly identified. Cucurbit breeders still rely on genetic maps, loci, and QTLs to describe genetic resistance for conventional breeding, but specific resistance genes and the mechanism of anthracnose resistance have not been characterized. The whole genome sequence available for watermelon and advancements in genomics, transcriptomics, proteomics, and metabolomics offer the opportunity to identify genes responsible for the anthracnose-resistant phenotype; however, these approaches are sensitive to noise from phenotyping and genotyping and do not always result in a clear candidate gene. The occurrence of races in *C. orbiculare* is an indication that pyramiding resistance will be required to ensure the durability of the trait and minimize the risk of new isolates overcoming individual resistance genes as has occurred with other cucurbit diseases [48,71]. Establishing the resistance gene repertoire in watermelon and characterizing the interactions of such proteins with pathogen proteins that result in a resistant phenotype will be needed to achieve durable anthracnose resistance in watermelon. Likewise, continued identification of new and improved resistance sources will remain a priority for breeding anthracnose resistance.

**Author Contributions:** Conceptualization, T.P., T.C.W., L.M.Q.-O. and S.M.; writing—original draft preparation, T.P., B.P.B. and E.C.; writing—review and editing, T.C.W., L.M.Q.-O. and S.M.; supervision, T.C.W. and S.M.; funding acquisition, S.M. All authors have read and agreed to the published version of the manuscript.

**Funding:** This research was supported by USDA Hatch Project TEX09665, Texas A&M AgriLife Vegetable Seed Grant, Texas A&M University Excellence Fellowship, and Texas A&M AgriLife Research Strategic Initiative Assistantship.

**Data Availability Statement:** No new data were created or analyzed in this study. Data sharing is not applicable to this article.

**Conflicts of Interest:** The authors declare no conflict of interest. The funders had no role in the design of the study; in the collection, analyses, or interpretation of data; in the writing of the manuscript; or in the decision to publish the results.

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
