# Peer review of "Recent Advances and Challenges in Management of Colletotrichum orbiculare, the Causal Agent of Watermelon Anthracnose"

_horticulturae, doi:10.3390/horticulturae9101132_

Round 1

Reviewer 1 Report

(1)It is recommended that the author present a functional morphology and life and disease cycle in the form of a simulation diagram of their abilities.

(2)Are the images of the article taken by the author himself or are they based on other website sources? If not, please indicate the source of the images. If so, please provide scale.

(3) It is recommended that the author present pathogenesis genes and effectors with table list in which include genes functions to watermelon anthracnose, references and so on.

Author Response

(1) It is recommended that the author present a functional morphology and life and disease cycle in the form of a simulation diagram of their abilities.

Response: Reviewer provided a good suggestion. However, this is a review but not a research manuscript, authors are not comfortable to draw a disease cycle as they had not conducted research on disease cycle.

(2) Are the images of the article taken by the author himself or are they based on other website sources? If not, please indicate the source of the images. If so, please provide scale.

Response: Thank you for the query and suggestion. These images were taken by the first author. Unfortunately, these did not have scale on them.

(3) It is recommended that the author present pathogenesis genes and effectors with table list in which include genes functions to watermelon anthracnose, references and so on.

Response: Thank you for the comment. There are not many effectors identified, hence tabulation might not be needed. We have mentioned and explained the pathogenesis genes and effectors in paragraphs.

Reviewer 2 Report

The article presents the detailed review of existing bibliography on the problem of watermelon anthracnose. The available data is carefully systematized starting from the history of the disease first description. Pathogen is described from different points of view: crops, susceptible to the disease; geographic distribution, taxonomy of the fungi; its morphology; symptoms of the disease; infection process; pathogen life cycle; measures of disease prevention, etc. It is important, that infection process is described in details both from the side of pathogen and host. Also the last advances in genetic analyses of fungi virulence and host resistance, including identification of resistance genes in the host and virulence genes of fungi and breeding of resistant varieties are presented.

The review will be useful for agronomists, geneticists, breeders, producers, and even consumers of watermelons and other cucurbits.      

At the same time I have some remarks:

1. Data of watermelon production (Page 1 line 33-34) have to be updated.

2. “Abstract” does not present all aspects of the topic, discussed in the article.

The article can be published after minor revision.

Author Response

  1. Data of watermelon production (Page 1 line 33-34) have to be updated.

Response: Thank you for the comment. We updated the data.

  1. “Abstract” does not present all aspects of the topic, discussed in the article.

Response: Thank you for this observation. We updated the abstract to reflect all the topics discussed later in the article.

Reviewer 3 Report

The work is a very good contribution since there are no reviews on this pathosystem. The manuscript is well thought out and organized. I found no drawbacks

Author Response

The work is a very good contribution since there are no reviews on this pathosystem. The manuscript is well thought out and organized. I found no drawbacks.

Response: The authors would like to thank you for your appreciation of our efforts. 

Reviewer 4 Report

A very well written review article. In my opinion, the only thing missing are photos showing Colletotrichum orbiculare on the medium and/or mycelial hyphae, spores (section 3.3 Fungal Morphology).

Author Response

A very well written review article. In my opinion, the only thing missing are photos showing Colletotrichum orbiculare on the medium and/or mycelial hyphae, spores (section 3.3 Fungal Morphology).

Response: We would like to highlight again that it is a review manuscript. Authors do not have photos of fungus to show fungal growth on the medium, hyphae, and spores with a scale.